# LEARN TO KNOW UNKNOWNS: A BIONIC MEMORY NETWORK FOR UNSUPERVISED ANOMALY DETECTION

## ABSTRACT

Is generalization always beneficial? Over-strong generalization induces the model insensitive to anomalies. Unsupervised anomaly detection requires only unlabeled non-anomalous data to learn and generalize normal patterns, which results in a modest reconstruction error when reconstructing normal instances and a significant reconstruction error when reconstructing anomalies. However, over-strong generalization leads to the indistinguishable reconstruction error of normal instances and anomalies, which means that the model well reconstructs the unknown anomalies, resulting in unnoticeable reconstruction error. Inspired by the cascade structure of the hippocampus and cortex in human brain memory, we proposed a re-representation memory network called Random Forgetting Twin Memory (RFTM) to decompose the latent space and introduce a configurable reintegration mechanism to suppress overgeneralization. RFTM shows striking brain-like memory characteristics, which enables the model to know what it does not know. RFTM has the convenience of a single line of code boosting at the model level without adding any additional extra loss terms at the loss function level. RFTM-based models have achieved state-of-the-art experimental results on different public benchmarks.

## 1 INTRODUCTION

Anomaly detection (AD) refers to the identification of deviant samples based on known rules, expectations, or distributions. It has been extensively studied in many fields requiring attention to rare events, such as healthcare(Šabić et al., 2021; Arabahmadi et al., 2022), financial fraud(Hilal et al., 2021; Sanober et al., 2021), and defect detection(Fu et al., 2022; Cui et al., 2022). The Machine learning-based approach is gradually adopted to deal with AD tasks with more and more impressive results. Supervised and unsupervised methods are two branches of machine learning-based AD(Omar et al., 2013). The former requires labeled data for model training, while the latter does not. Due to the occurrence of anomalies being a small probability event, the supervised approach is restricted by the extremely imbalanced samples, which means there are not sufficient labeled anomaly samples for the model to learn. In contrast, the unsupervised approach does not require anomaly samples. Unsupervised anomaly detection (UAD) only needs to fit the unlabeled normal samples to learn the normal patterns, so it can identify the anomaly when the instance deviates from the normal patterns. Therefore, UAD has attracted extensive research interest due to its label-free characteristic. Reconstruction-based frameworks have achieved excellent performance in image(Li et al., 2021; Schneider et al., 2022), video(Deepak et al., 2021; Chang et al., 2022), and time series(Thill et al., 2021; Kieu et al., 2022) AD tasks in recent years as a classic UAD paradigm. But at the same time, some studies(Gong et al., 2019; Park et al., 2020) show that reconstruction-based AD has an overgeneralization problem (OGP).

Generalization is not always a good thing. Over-strong generalization in UAD leads to the failure of the reconstruction model to detect anomalies. The OGP was first observed in Figure 1 of paper Zong et al. (2018). Gong et al. (2019) first briefly demonstrated and reported this problem in a one-class classification (OC) case. We give the first formal description of OGP in a broad sense in the following and demonstrate it in a scenario that is not limited to OC. The framework of reconstruction-based AD is shown in Figure 1b. The model is first trained by minimizing the reconstruction error of normal samples in the training phase, and then the trained model measures the anomaly degree

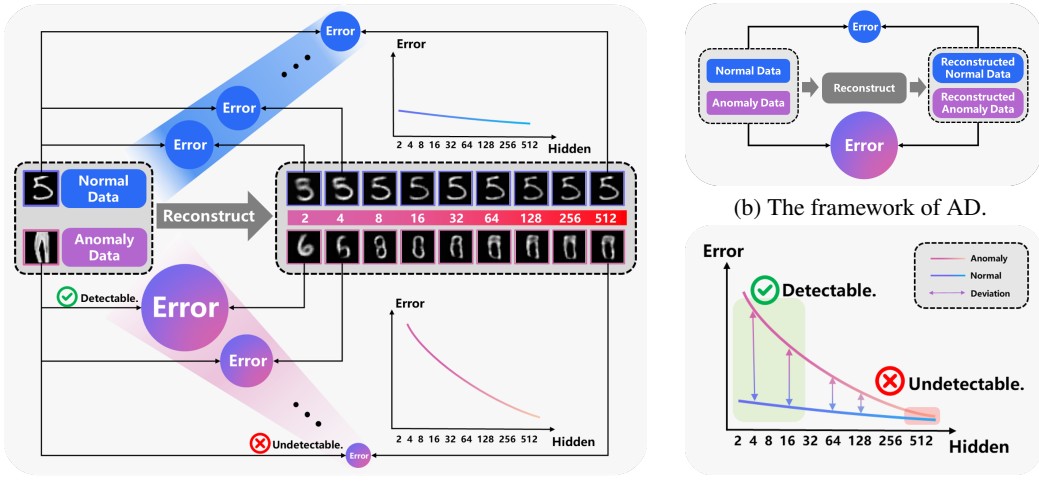

(a) The visual illustration of OGP in reconstruction.

(b) The framework of AD.

(c) The collapse of IIA.

Figure 1: OGP is demonstrated in the inference stage by using reconstruction models with the different number of bottleneck neurons trained on MNIST. The increasing number of bottleneck neurons from 2 to 512 stands for the increasing complexity of the model. The images from MNIST are taken as normal, and the input from Fashion-MNIST is taken as anomalous. It can be seen that the reconstruction error gently decreases with the increase of model complexity for the normal class handwritten digit 5. The reconstruction error decreases rapidly with the increase of model complexity for the trousers image from Fashion-MNIST. The model trained on MNIST could reconstruct the data from Fashion-MNIST well when the complexity increased to a certain extent. The deviation decreases with the increase of model complexity, which leads to the anomalies undetectable.

of unseen instances by the reconstruction error in the inference phase. The reconstruction error directly reflects the deviation between the unknown samples and the learned normal patterns. The larger the reconstruction error is, the higher the corresponding anomaly degree is. However, this framework has an implicit ideal assumption (IIA) that the model trained on the normal samples cannot reconstruct the anomaly samples well. The model generalization is enhanced with the increase of model complexity which leads to the input data indiscriminately being reconstructed too well as shown in Figure 1a. The input data containing both normal and anomaly samples will output the reconstruction error with an indistinguishable difference in the inference phase due to the excessive generalization of the model, resulting in the collapse of IIA as shown in Figure 1c.

The OGP is particularly prominent and challenging in unsupervised semantic anomaly detection (USAD). The example shown in Figure 1a is a routine setting, that is, using one dataset as normal and the other as anomalous. Different from the routine setting, the motivation proposed by Ahmed & Courville (2020) emphasizes that anomalies should be at the semantic level, which recommends that hold out one class as anomalous and the rest classes as normal within the same dataset. The main challenges of USAD are as follows. First, the labels of normal classes are not accessible, which makes it impossible to train the classifier with supervised labels to obtain tight bounds that can describe normal patterns. Second, the number of classes of the normal pattern is unknown, which means that the OGP is more likely to occur in this multimodal case because more unlabeled classes of data lead to better generalization. Third, the anomalies are at the semantic level and invisible, which means that the OGP can easily occur if the bounds on the normal pattern are not tight enough. Finally, the OGP can be alleviated by reducing the complexity of the reconstruction model from the perspective of its origin. However, there may be no model with optimal complexity that can both well generalize normal samples and fail to overgeneralize anomaly samples to satisfy IIA. Besides, multiple training searches are computationally intolerable even if the optimal solution exists.

In recent years, a part of the work has started to pay attention and try to solve the above problems. MemAE(Gong et al., 2019) proposed a memory module that stores a certain amount of prototype of latent information and re-represents the latent space with a sparse attention mechanism. MNAD(Park et al., 2020) introduces a transformation matrix to remap the latent space, using

a KMeans-like approach to allow the memory module to be updated independently. DAAD(Hou et al., 2021) embeds the memory module(Gong et al., 2019) in more than one location of the network. Unfortunately, all the above works introduce one or more penalty terms to induce the model to be functional. The adjustment of the penalty term coefficients largely affects the effectiveness of the model. Besides, most of the experiments were not conducted based on the USAD setup.

We proposed a novel re-representation neural network called Random Forgetting Twin Memory (RFTM) and introduced a configurable reintegration mechanism to solve the OGP in USAD. Specifically, RFTM uses limited prototypes to record and re-represent the latent representation, which makes suppression of overgeneralization feasible. Beyond that, RFTM can be trained end-to-end without introducing any additional penalty terms on top of the original task loss function. It is important to note that the RFTM is a well-packaged module, which allows a single line of code in a project deployment to achieve performance gains for OGP without changing the original model. We conducted experiments and achieved state-of-the-art performance on different kinds of public benchmarks. The main contributions of this paper are as follows:

- **RFTM networks**: A novel memory network for reconstruction-based AD with a bionic cascade structure and a random forgetting mechanism is proposed to suppress overgeneralization.

- **Plug-and-Play**: RFTM is a plug-and-play network that requires only one line of code to be added and does not require any modification of the original loss in the original project to obtain improved results for OGP.

- **Bionic properties**: We demonstrated the process of RFTM training, which is surprisingly similar to the process of memory formation in the human brain. We experimentally demonstrated the human brain's similarity between RFTM and hippocampus-cortex structures.

- **Effectiveness**: We performed extensive experiments on various datasets and achieved state-of-the-art performance to demonstrate the effectiveness of the RFTM.

## 2 RELATED WORK

**Unsupervised Anomaly Detection.** Reconstruction-based methods learn a model that is optimized to well-reconstruct normal data instances, thereby aiming to detect anomalies by failing to accurately reconstruct them under the learned model(Ruff et al., 2021). Autoencoders are naturally chosen as the model framework for reconstruction-based AD(Hou et al., 2021). The reconstruction error is usually considered to be small on the normal training set and large on the anomaly test set(Hasan et al., 2016; Zhao et al., 2017; Zong et al., 2018).

**Memory Bank.** The OGP has progressed to some extent in the field of video surveillance since the memory module was proposed by Gong et al. (2019). At the model structure level, the memory module provides an additional trainable matrix between the encoder and decoder for storing prototypes and introduces an attention mechanism to reintegrate prototypes. At the loss function level, the memory module suppresses overgeneralization by adding regular terms that encourage attention weights sparsity. The success of the memory module drew the attention of scholars to improve it. MNAD(Park et al., 2020)reconstructs latent information from the clustering perspective of memory read/write updates. At the model structure level, MNAD draws on ideas from the K-Menas algorithm. Each row of the trainable matrix is considered as a clustering center, which is momentum updated by the neighbor information. At the loss function level, separation loss and compact loss are introduced to control the inter-class and intra-class distance centered on the prototype. Trust-MAE(Tan et al., 2021) based on MNAD further introduces the trust region to suppress overgeneralization. Six regular terms are introduced to jointly constrain the loss function. DAAD(Hou et al., 2021) embeds memory module(Gong et al., 2019) in all concatenate layers of Unet. The adversarial and alignment loss regular terms are introduce to train. The main improvements of the memory module are as follows. First, the regular term is designed to work with the memory module to suppress overgeneralization. Second, the memory module is updated autonomously in a non-gradient style similar to clustering. Third, the memory module is embedded in different positions of the model. However, the solution to new problems of adjusting the coefficients of the regular terms, the unknown prior capacity setup of the memory module, and the choice of the embedding position of the model remain explored.

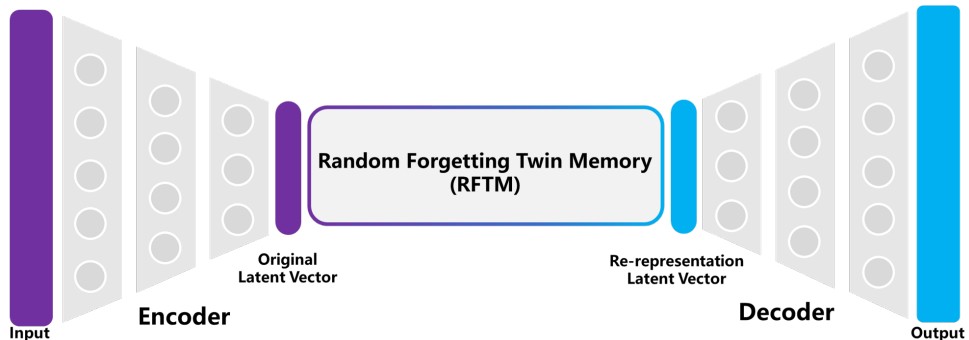

(a) The RFTM-based reconstruction framework.

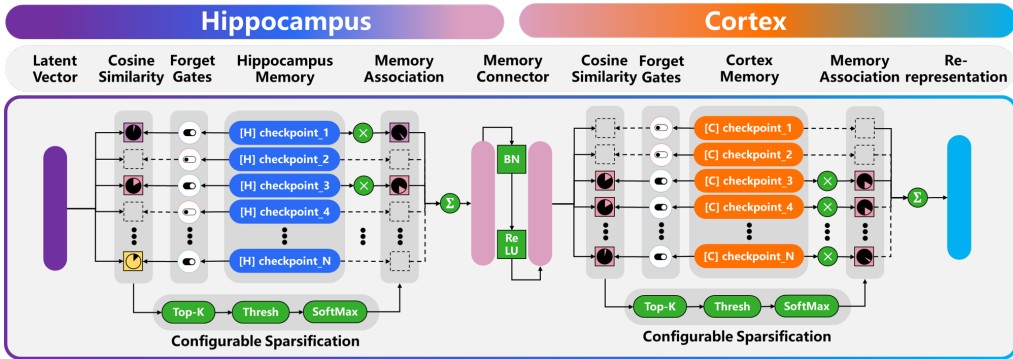

(b) The details of the RFTM.

Figure 2: Overview of RFTM-AE.

## 3 METHOD

### 3.1 OVERVIEW

The reconstruction framework equipped with RFTM is shown in Figure 2a. The model consists of three parts: encoder, decoder, and RFTM. During the inference phase, the encoder $f_{\theta_E} : \mathbb{R}^D \mapsto \mathbb{R}^L$ firstly makes a nonlinear transform to the input data $x \in \mathbb{R}^D$ and represents them in the latent space $\mathbb{R}^L$ with parameter $\theta_E$. Hence, the latent representation can be written by $h = f_{\theta_E}(x) \in \mathbb{R}^L$. Then the RFTM $\xi_{\theta_M} : \mathbb{R}^L \mapsto \mathbb{R}^L$ re-represent the latent representation to get $\widetilde{h} = \xi_{\theta_M}(h)$. After that, the decoder $\varphi_{\theta_D} : \mathbb{R}^L \mapsto \mathbb{R}^D$ receives re-representation $\widetilde{h}$ from the RFTM and, as opposed to the encoder, reconstructs it to the original input data with parameter $\theta_D$ as closely as possible. In the end, the anomaly intensity $\kappa$ of the unseen input sample is measured by reconstruction error. Equation (1) gives the formal description of the inference phase.

$$\kappa = \|\varphi_{\theta_D}(\xi_{\theta_M}(f_{\theta_E}(x))) - x\| \tag{1}$$

During the training phase, the model minimizes reconstruction error and uses BP(Rumelhart et al., 1986) algorithm to update all parameters including RFTM. At the same time, the RFTM learned representative patterns as prototype $\theta_M$, which is demonstrated in Section 3.2. The optimization problem is described by Equation (2).

$$\min_{\theta_E, \theta_M, \theta_D} \|\varphi_{\theta_D}(\xi_{\theta_M}(f_{\theta_E}(x))) - x\|_2^2 \tag{2}$$

Without loss of generality, it should be noted that the proposed RFTM network is suitable for all models that follow the autoencoder paradigm and works for the most shape of data. That is to say, $f$ and $\varphi$ mentioned above can be neural network operators such as CNN(Krizhevsky et al., 2012), RNN(Zaremba et al., 2014), GCN(Kipf & Welling, 2016) and their variations, and the space $\mathbb{R}^D$ to which the input data belongs can be multidimensional time series, images or videos, etc.

## 3.2 RANDOM FORGETTING TWIN MEMORY

### 3.2.1 BASIC UNIT

**Memory.** Consider a trainable matrix $M \in \mathbb{R}^{N \times L'}$, where $N$ is the memory capacity and $L'$ is the flattened shape of $L$. Let $m_i$ be the $i$-th row of the matrix $M$ denoting the $i$-th prototype checkpoint of the memory.

**Cosine Similarity.** The similarity $s_i = hm_i/|h||m_i| \in [-1, 1]$ between the latent representation $h$ and any of the prototype checkpoint $m_i$ can be calculated. Let vector $s = (s_1, s_1, ..., s_N) \in \mathbb{R}^N$ be the association weights. Let $s = S(h, M)$, where $S$ is the operator of the vector $s$ with respect to $h$ and $M$.

**Random Forgetting.** The forget gates part is a binary vector $\rho_\nu \in \{0, 1\}^N$ controlled by the forgetting rate $\nu \in [0, 1]$, where the $\nu N$ elements are randomly and non-repeatedly selected to be set to one and the rest to zero.

**Configurable Sparsification.** The configurable sparsification consists of three functions, the Top-K function, the threshold function, and the softmax function. The Top-K function $T_k(x)$ sets the index of the first $k$ largest component of each component of $x$ to one and the rest to zero. The threshold function $\tau_\alpha(x)$ sets the index of all components of $x$ greater than $\alpha$ to be one and the rest to be zero. The $\sigma(x)$ is softmax function. Note that function $\pi$ is a compression function that removes the component of the input vector or matrix that is zero or an integer row that is all zero and outputs a vector or matrix of the original order consisting of the remaining components or rows. In particular, we always want to use similar memory checkpoints to re-represent, so let $\alpha = 0$.

**Memory Association.** The output of the basic unit is obtained by linear summing of the output vector of the above pipeline and the prototype memory.

$$h_{out} = \sigma(\tau_0(\pi(T_k(\rho_\nu \odot S(h_{in}, M)))))\pi(\rho_\nu e^T \odot M) \tag{3}$$

Where $e = \{1\}^{L'}$ is the broadcast vector whose components are all one, which is used to add a mask to the forgotten prototype memory.

### 3.2.2 TWIN MEMORY

**Memory Connector.** The memory connector consists of BN and LeakyReLU. For convenience, the BN-LeakyReLU block is noted as the operator $P$.

**Cascade Structure.** The output $h \in \mathbb{R}^L$ of the encoder will first perform a flattening operation to shape $L'$ and will be restored to its original shape after RFTM processing. Without loss of generality, let $L = b \times h \times w \times c$ for the example of latent space of image data, then $L' = b \times hwc$. RFTM consists of a pair basic unit of $\zeta_{M_H} : \mathbb{R}^{L'} \mapsto \mathbb{R}^{L'}$ and $\mu_{M_C} : \mathbb{R}^{L'} \mapsto \mathbb{R}^{L'}$, where $\theta_M = \{M_H, M_C\}$. The former is the hippocampus mapping and the latter is the cortex mapping. They share an identical architecture in Section 3.2.1 but different hyperparameters $\{N, k, \nu\}$ connected by a memory connector as shown in Figure 2a.

**RFTM.** The formal description of RFTM is shown in the following Equation 4 and Equation 5.

$$h' = \sigma(\tau_0(\pi(T_{k_H}(\rho_{\nu_H} \odot S(h, M_H)))))\pi(\rho_{\nu_H} e^T \odot M_H) \tag{4}$$

$$h''' = \sigma(\tau_0(\pi(T_{k_C}(\rho_{\nu_C} \odot S(h'', M_C)))))\pi(\rho_{\nu_C} e^T \odot M_C) \tag{5}$$

Where $h'' = P(h')$. $\widetilde{h} = P(h''')$ is treated as the output of RFTM.

## 4 EXPERIMENTS

### 4.1 DATASETS AND EVALUATION

We selected three benchmarks as follows, MNIST(Deng, 2012),FMINST(Xiao et al., 2017), and CIFAR10(Krizhevsky et al., 2009). All datasets follow the USAD settings. Each class in the dataset

Table 1: The numerical results.

| | AUROC | | | | AUPRC | | | | F1 | | | |
|---|---|---|---|---|---|---|---|---|---|---|---|---|
| **MNIST** | AE | MemAE | MNAD | Ours | AE | MemAE | MNAD | Ours | AE | MemAE | MNAD | Ours |
| 0 | 79.16±0.30 | 86.16±1.20 | 79.88±0.33 | **96.98±0.08** | 25.83±0.69 | 39.98±3.23 | 26.50±0.61 | **76.14±1.00** | 36.52±0.52 | 44.85±1.38 | 36.63±0.59 | **73.34±0.61** |
| 1 | 12.45±0.94 | 17.16±2.93 | 16.90±1.43 | **40.04±5.24** | 6.27±0.04 | 6.57±0.16 | 6.51±0.08 | **8.70±0.76** | 20.39±0.01 | 20.42±0.05 | 20.39±0.01 | **21.46±0.55** |
| 2 | 86.20±0.22 | 93.80±1.35 | 86.63±0.64 | **97.05±0.26** | 44.93±0.52 | 67.98±6.65 | 46.16±1.21 | **81.23±1.55** | 48.47±0.78 | 65.39±4.26 | 49.24±1.17 | **74.73±0.78** |
| 3 | 64.68±0.10 | 82.50±3.12 | 66.62±0.41 | **91.07±0.21** | 16.84±0.22 | 31.06±4.58 | 17.31±0.27 | **48.92±1.37** | 22.98±0.04 | 38.26±3.51 | 23.91±0.14 | **53.55±0.93** |
| 4 | 59.13±0.45 | 77.04±3.14 | 65.29±4.48 | **82.99±0.27** | 13.24±0.25 | 25.84±3.25 | 16.44±2.61 | **33.62±0.90** | 20.40±0.11 | 33.11±2.57 | 23.85±3.06 | **39.24±0.52** |
| 5 | 71.34±0.15 | 81.16±2.69 | 72.13±0.62 | **92.51±0.27** | 16.26±0.13 | 25.80±3.49 | 17.08±0.40 | **53.47±1.59** | 25.83±0.26 | 35.61±3.29 | 26.72±0.46 | **54.84±1.10** |
| 6 | 84.92±0.30 | 89.64±0.52 | 85.61±1.49 | **94.16±0.22** | 41.85±0.83 | 54.59±1.00 | 42.96±2.49 | **67.72±1.26** | 47.46±0.77 | 56.01±1.21 | 47.64±2.18 | **63.27±0.74** |
| 7 | 61.91±0.85 | 61.20±3.09 | 67.45±2.13 | **79.90±1.21** | 19.13±0.69 | 15.15±1.25 | 23.09±1.66 | **31.80±2.43** | 22.78±0.75 | 21.73±1.32 | 27.69±1.39 | **38.65±2.30** |
| 8 | 67.84±0.16 | 91.06±0.54 | 75.90±1.58 | **93.58±0.30** | 14.98±0.06 | 49.03±2.02 | 20.97±1.42 | **56.36±1.99** | 24.98±0.15 | 51.88±1.52 | 29.96±1.36 | **58.53±1.42** |
| 9 | 44.38±1.70 | 53.54±3.41 | 51.29±2.24 | **70.00±0.53** | 8.82±0.31 | 10.31±0.76 | 9.98±0.49 | **17.55±0.52** | 19.77±0.05 | 21.38±1.02 | 20.68±0.30 | **26.31±0.22** |
| AVG | 63.20±0.52 | 73.33±2.20 | 66.77±1.53 | **83.83±0.86** | 20.81±0.37 | 32.63±2.64 | 22.70±1.12 | **47.55±1.34** | 28.96±0.34 | 38.86±2.01 | 30.67±1.07 | **50.39±0.92** |
| **Fashion** | AE | MemAE | MNAD | Ours | AE | MemAE | MNAD | Ours | AE | MemAE | MNAD | Ours |
| T-shirt | 57.03±0.24 | 55.66±0.92 | 57.15±1.99 | **60.07±0.13** | 11.57±0.03 | 11.62±0.19 | 11.67±0.72 | **13.59±0.19** | 20.80±0.12 | 20.21±0.37 | 20.95±0.70 | **21.32±0.08** |
| Trouser | 71.03±1.25 | 88.78±0.48 | 70.06±2.11 | **85.05±0.07** | 15.37±0.50 | 37.73±1.33 | 14.76±1.15 | **31.07±0.29** | 28.67±0.96 | **50.6±0.92** | 28.45±0.95 | 43.17±0.40 |
| Pullover | 43.64±0.25 | 52.75±0.41 | 45.19±1.13 | **62.84±0.26** | 8.91±0.03 | 10.25±0.09 | 9.03±0.20 | **13.52±0.03** | 18.81±0.14 | 20.24±0.09 | 18.95±0.12 | **22.48±0.20** |
| Dress | 61.30±0.97 | 67.52±1.16 | 63.47±1.87 | **70.38±0.92** | 12.32±0.28 | 14.27±0.44 | 12.90±0.73 | **15.89±0.47** | 20.33±0.01 | 25.79±0.63 | 24.1±0.72 | **27.51±0.67** |
| Coat | 48.93±0.16 | 52.32±1.50 | 48.74±0.21 | **60.95±0.50** | 8.94±0.03 | 9.64±0.26 | 8.87±0.05 | **12.10±0.11** | 20.33±0.01 | 20.81±0.47 | 20.29±0.07 | **22.75±0.15** |
| Sandal | **92.28±0.46** | 89.25±0.34 | 91.85±0.92 | 84.89±0.62 | 49.03±1.46 | 44.75±0.55 | **49.71±2.24** | 37.95±0.25 | 55.15±1.43 | 47.93±0.44 | 53.68±2.73 | 40.87±1.14 |
| Shirt | 51.15±0.52 | 52.63±0.96 | 51.20±0.53 | **55.66±0.77** | **11.34±0.15** | 11.08±0.22 | 11.12±0.13 | 11.28±0.21 | 19.05±0.08 | 19.45±0.18 | 19.18±0.06 | **20.13±0.21** |
| Sneaker | **64.85±0.4** | 60.66±1.26 | 61.75±2.77 | 61.68±1.81 | **12.33±0.14** | 11.20±0.32 | 11.45±0.73 | 11.61±0.51 | **26.77±0.19** | 24.31±0.65 | 25.46±1.18 | 23.91±0.82 |
| Bag | 95.56±0.17 | 94.84±0.36 | **96.13±0.18** | 95.56±0.11 | 66.06±1.29 | 58.42±2.1 | **69.47±1.26** | 66.84±0.41 | 70.06±0.78 | 65.35±1.10 | **71.45±0.17** | 68.53±0.27 |
| Ankle boot | 83.41±0.64 | 82.22±0.65 | **83.78±0.61** | 79.63±0.57 | 27.69±0.54 | 27.30±0.81 | **28.93±1.36** | 28.76±0.91 | 38.82±0.83 | 36.98±0.75 | **39.36±0.65** | 37.31±0.32 |
| AVG | 66.92±0.51 | 69.66±0.80 | 66.93±1.23 | **71.67±0.58** | 22.36±0.45 | 23.63±0.63 | 22.79±0.86 | **24.26±0.34** | 32.16±0.49 | **33.17±0.56** | 32.19±0.73 | 32.80±0.43 |
| **CIFAR10** | AE | MemAE | MNAD | Ours | AE | MemAE | MNAD | Ours | AE | MemAE | MNAD | Ours |
| airplane | 38.60±0.13 | 37.62±0.25 | 38.41±0.61 | **42.77±0.03** | 8.08±0.01 | 7.67±0.08 | 7.93±0.09 | **8.58±0.03** | 18.18±0.01 | 18.18±0.01 | 18.18±0.01 | **18.19±0.01** |
| automobile | 69.00±0.75 | **74.48±0.03** | 71.05±0.35 | 72.21±0.37 | 19.47±0.54 | 21.11±0.03 | 20.01±0.52 | 18.02±0.22 | 25.62±0.50 | **29.99±0.06** | 27.24±0.37 | 28.52±0.17 |
| bird | **43.86±0.12** | 37.81±0.17 | 40.86±0.20 | 38.62±0.19 | **8.66±0.02** | 7.65±0.04 | 8.14±0.03 | 8.11±0.05 | **18.28±0.01** | 18.26±0.01 | 18.25±0.01 | 18.19±0.01 |
| cat | 42.22±0.07 | 45.89±0.10 | 43.01±0.08 | **56.35±0.03** | 8.17±0.01 | 8.73±0.02 | 8.24±0.02 | **12.14±0.08** | 18.31±0.01 | 18.30±0.03 | 18.25±0.01 | **19.73±0.12** |
| dog | **45.80±0.36** | 37.82±0.34 | 42.26±0.42 | 30.44±0.18 | **8.60±0.07** | 7.44±0.05 | 8.03±0.08 | 6.63±0.01 | **18.35±0.03** | 18.20±0.01 | 18.20±0.01 | 18.18±0.01 |
| deer | 41.63±0.07 | 46.96±0.15 | 42.21±0.18 | **59.23±0.37** | 8.05±0.02 | 8.82±0.02 | 8.06±0.03 | **13.35±0.29** | 18.30±0.02 | 18.60±0.03 | 18.32±0.02 | **20.89±0.07** |
| dog | **65.39±0.14** | 59.55±0.50 | 63.76±0.61 | 38.53±0.03 | **15.85±0.1** | 13.60±0.32 | 15.31±0.31 | 7.56±0.01 | **24.84±0.11** | 21.23±0.28 | 23.71±0.37 | 18.31±0.01 |
| frog | 55.45±0.39 | 59.79±0.16 | 57.32±0.21 | **61.48±0.23** | 10.50±0.11 | 12.25±0.06 | 11.10±0.05 | **13.79±0.15** | 20.27±0.04 | 21.51±0.06 | 20.63±0.05 | **21.74±0.12** |
| hourse | 39.55±0.10 | 39.51±0.22 | 39.33±0.45 | **44.03±0.02** | 7.72±0.01 | 7.72±0.03 | 7.67±0.06 | **8.48±0.01** | 18.18±0.01 | 18.18±0.01 | 18.18±0.01 | **18.20±0.01** |
| sheep | 65.61±0.18 | 71.00±0.24 | 69.27±0.57 | **71.14±0.26** | 16.10±0.11 | 19.51±0.09 | **17.97±0.33** | 17.79±0.14 | 24.31±0.14 | 28.17±0.28 | 26.73±0.49 | **27.69±0.35** |
| AVG | 50.71±0.23 | 51.04±0.22 | 50.75±0.37 | **51.48±0.20** | 11.12±0.1 | **11.45±0.07** | 11.25±0.15 | 11.44±0.10 | 20.46±0.09 | **21.06±0.08** | 20.77±0.13 | 20.96±0.08 |

was selected in turn as the anomalous class and the remaining as the normal class, thus constructing multiple sub-experiments under one dataset. 10% of the training data is split into validation sets according to the random seed 2022. In the training and validation sets, the anomalies are removed and only normal data are left. The label information of all normal classes is also removed to satisfy the USAD setting. In the test set, all normal data are labeled as negative, and anomalies are labeled as positive. AUROC, AUPRC, and F1 were used as evaluations for this experiment, where AUPRC can better reflect the performance of the model compared to AUROC in the unbalanced case.

## 4.2 IMPLANTATION DETAILS

To facilitate the description, we make the following notation convention. FC(a,b) denotes Linear-BN-LeakyReLU block, where a,b is number of input and output channels. Conv(a,b,k,s) denotes Conv-BN-LeakyReLU block, where (a,b,k,s) denotes input channels, output channels, kernel size, and strides, respectively. DeConv(a,b,k,s) denotes the ConvTranspose-BN-LeakyReLU block, where (a,b,k,s) has the same meaning as above. We use the same autoencoder in MNIST and Fahion-MNIST experiments, i.e., FC1(784,512), FC2(512,256), FC3(256,128), FC4(128,256), FC5(256,512), FC6(512,784). Because the CIFAR10 dataset is more informative with RGB color images, we use a convolutional autoencoder, i.e., Conv1(3,64,3,2), Conv2(64,128,3,2), Conv3(128,256,3,2), Conv4(256,512,3,2), DeConv5(512,256,3,2), DeConv6(256,128,3,2), DeConv7(128,64,3,2), DeConv8(64,3,3,2). Note that since MNAD requires a residual structure at the bottleneck, the first layer of his decoder is twice as large as the other models, namely: FC4(128+128,256) and DeConv5(512+512,256,3,2).

All parallel comparison models in Section 4 use the same encoder, decoder, random seed, training set, test set, and validation set. All experimental training procedures use the same batch size of 256, Adam optimizer with a 1e-3 learning rate, and early stopping with patience of 10. The hyperparameters $\{N, k, \nu\}$ for the twin memory is $\{50, 15, 0.1\}$ and $\{1000, 5, 0.5\}$. We boosted the hippocampus and cortex memory capacity to 3000 in the CIFAR10 experiment because color pictures are more informative. Vanilla AE, MemAE(Gong et al., 2019), MNAD(Park et al., 2020) were used as comparison models. The random seeds for the training process, validation set splitting, and inference evaluation process is all the same to ensure absolute fairness and achieve targeted comparisons for RFTM effectiveness. All experiments were performed in three replicate runs with random seeds 1997, 2015, and 2022 respectively.

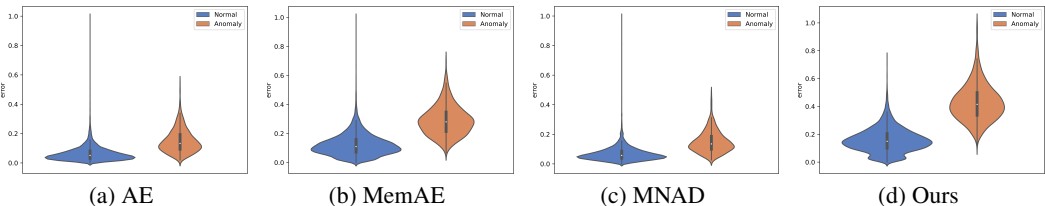

Figure 3: A demonstration of overgeneralization suppression.

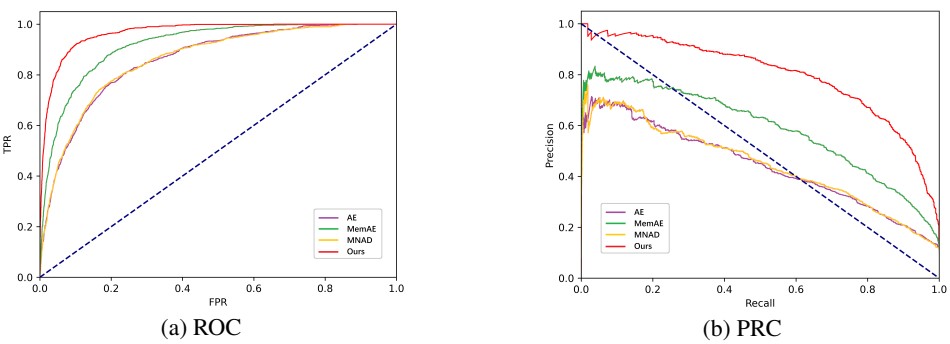

Figure 4: Curves of different models on two metrics.

### 4.3 MAIN RESULTS

We counted the results of the comparison tests in Table 1. Each row in the table represents a different metric for different models on that test set with that class as anomalous and the rest of the classes as normal. We use bold font to indicate the best results for each experiment under each metric. We achieve almost 20% improvement in the average AUROC metric on the MNIST dataset. We also achieve state-of-the-art results on Fashion-MNIST and CIFAR10 datasets, which suggests that RFTM has the ability to significantly inhibit OGP under USAD settings.

To better demonstrate the ability of RFTM to discriminate between normal and anomaly samples, we choose one of the experiments to normalize the reconstruction errors of all samples in the test set and draw the distribution according to the test labels as shown in Figure 3. Orange represents the reconstruction error distribution of anomaly class and blue represents the reconstruction error distribution of normal class. Our method compared to others produces a significant distance between the reconstruction error distributions of normal and anomaly as shown in Figure 3d, which shows that RFTM creates a significant distinction over other models in the USAD setting. The previous results can be verified by observing the ROC curve as shown in Figure 4a.

It is worth noting that the reconstruction error distribution of normal samples is relatively short-tailed in the output of the RFTM-based model, indicating that the model has a lower false alarm rate. By plotting the precision-recall curve (PRC) as shown in Figure 4b, it can be observed that RFTM always maintains a high precision for the same recall thus verifying the low false alarm rate due to the short-tailed distribution of normal reconstruction errors.

### 4.4 ABLATION STUDY

In this section, we carry out ablation experiments focusing on the influence of different basic units in RFTM. We randomly selected the experiments of Fashion-MNIST for Coat anomaly and performed the ablation experiments with fixed random seed 1997 as shown in Table 2. The performance of the original RFTM in the last row is used as a comparison criterion. The three rightmost columns represent the reduction of the ablation experiment relative to the performance of the original RFTM, where the largest, second largest, and third largest reductions are indicated in black, red, and blue, respectively. It can be seen that cortex memory has the largest effect on model performance. The hippocampus memory has the second largest effect. In addition, the active forgetting mechanism was shown to be effective.

Table 2: Results of ablation experiments on RFTM.

| Basic Unit Ablation | | | | | Evaluation | | | Reduction | | |
|---|---|---|---|---|---|---|---|---|---|---|
| Forget Gates | Sparsification | Connector | H-Memory | C-Memory | AUROC | AUPRC | F1 | AUROC↓ | AUPRC↓ | F1↓ |
| × | ✓ | ✓ | ✓ | ✓ | 59.60 | 11.66 | 22.24 | **2.01** | **0.60** | **0.72** |
| ✓ | × | ✓ | ✓ | ✓ | 60.39 | 12.83 | 23.46 | 1.22 | -0.57 | -0.50 |
| ✓ | ✓ | × | ✓ | ✓ | 61.10 | 12.21 | 22.34 | 0.51 | 0.05 | 0.62 |
| ✓ | ✓ | ✓ | × | ✓ | 58.79 | 11.46 | 21.88 | **2.82** | **0.80** | **1.08** |
| ✓ | ✓ | ✓ | ✓ | × | 55.09 | 10.31 | 21.20 | **6.52** | **1.95** | **1.76** |
| ✓ | ✓ | ✓ | ✓ | ✓ | 61.61 | 12.26 | 22.96 | 0.00 | 0.00 | 0.00 |

(a) Input     (b) AE     (c) MemAE     (d) MNAD     (e) Ours

Figure 5: Comparison of the input images in the test set with the reconstructed images of different models.

## 4.5 BIONIC FEATURES

### 4.5.1 WHAT DOES RFTM DO?

RFTM learned to know the unknown by suppressing overgeneralization. We selected the first 50 samples in the test set as shown in Figure 5a. These samples were mixed with the semantic anomaly handwritten digit 2. We compare the reconstructed results of the four models for these 50 samples. The top of the image is marked with the test ground truth, where 0 means negative normal and 1 means positive anomaly. The output of the parallel comparison model proves that it falls into the overgeneralization trap as shown in Figure [5b,5c,5d]. This means that AE, MemAE, and MNAD all learn how to reconstruct well the handwritten digit 2 that does not exist in the training set. The reconstruction results in the overgeneralization case have imperceptible differences at the naked eye level, however, AD estimates the anomaly degree of the sample by evaluating the reconstruction error, which indicates that the high reconstruction error in the overgeneralization case results from the image noise that is not visible at the naked eye level. This case is not interpretable and risks the failure of AD as the model parameter capacity is further increased.

The reconstruction results of the RFTM-based model satisfy the IIA. For all anomalous samples, the reconstructed results of the RFTM-based model exhibit boundary tightness and interpretability. On the one hand, RFTM does not reconstruct semantics that does not exist outside the training set, and on the other hand, RFTM only reconstructs semantics that already exists in the training set and can be understood by the naked eye. Compared to other models that identify anomalies by noise in the overgeneralization case, RFTM has a larger reconstruction error and better interpretation. RFTM reconstructs the anomalous samples as handwritten numbers 0, 8, 3, 4, and 5 as shown in Figure 5e. This shows that the model only knows what it knows. In other words, RFTM gives the model the ability to know what it does not know.

### 4.5.2 HOW DOES RFTM WORK?

In order to interpret RFTM from the representation space, we have done the following visualization. The outputs of the encoder, hippocampus memory, and cortex memory were selected to perform the representation visualization in the test set as shown in Figure 7. Purple color indicates normal samples and yellow color indicates anomalies. The encoder always tends to construct a space with cavities as shown in Figure 7a, which are used to accommodate anomaly representations. However, the presence of cavities in the representation space causes the model to learn the anomalous representation spontaneously, providing the basis for the decoder to reconstruct the anomalous pattern well (OGP).

Based on the encoder output space, hippocampus memory tries to construct a denser and cavity-free space as shown in Figure 7b. To produce large reconstruction errors for anomalous samples,

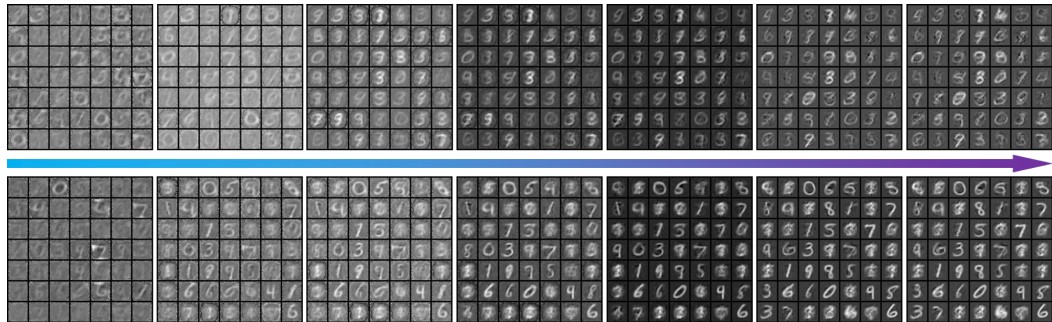

Figure 6: We tracked the update of 49 randomly selected memory checkpoints in hippocampus memory and cortex memory, respectively. Above the arrow is hippocampus memory and below is cortex memory.

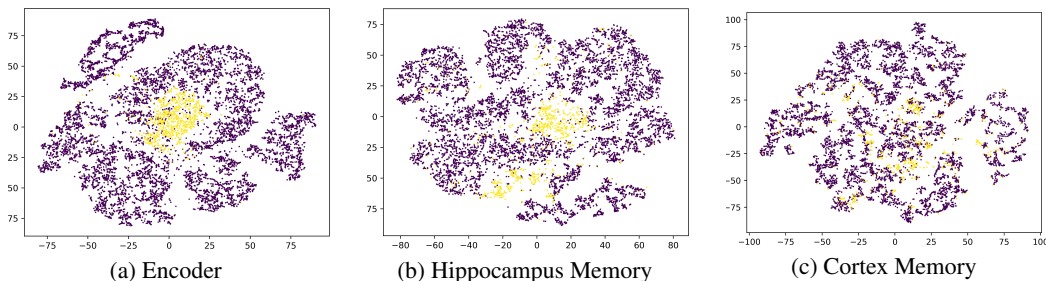

(a) Encoder        (b) Hippocampus Memory        (c) Cortex Memory

Figure 7: 2D visualization of different stage representation output by tSNE.

the model needs to have a representation space that is dense enough that the anomalies are always forced to be embedded in the distribution of normal representations. The re-representation space of the cortex memory achieves enough denseness mentioned before. Figure 7c shows that the lack of cavities forces the anomalies to be distributed within the boundaries of the normal pattern. We illustrate the memory checkpoint generation in the RFTM training process in Figure 6 to intuitively verifies the above explanation. It can be observed that the prototype memory checkpoints used for memory association gradually learn representative normal representations in the training set. The new space constructed based on these representative checkpoints will be denser and cavity-free, forcing the anomaly representation to be re-represented to known normal patterns as shown in Figure 5e.

## 5    CONCLUSION AND FUTURE WORK

We proposed the RFTM for UAD inspired by the bionics of the hippocampus-cortex cascade structure, which has yielded state-of-the-art results in experiments. The RFTM has the following advantages. First, it can effectively inhibit overgeneralization to know the unknown anomalies. Second, it is a network designed purely from a structure perspective without adding additional loss terms, which makes it plug-and-play with a single line of code. Third, RFTM has bionic properties similar to those of the human brain such as active forgetting, memory association, hippocampus memory, and cortex memory, which provides potential new perspectives on the interpretability of unsupervised data reconstruction. RFTM has values that deserve to be studied and explored in more depth, such as the relationship between RFTM and generative networks, other variants of RFTM, the application of RFTM in multivariate time series analysis, etc.

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
