# OpenReview forum: "Learn to Know Unknowns: A Bionic Memory Network for Unsupervised Anomaly Detection"
_ICLR.cc/2023/Conference — Submitted to ICLR 2023_

### Official Review · Reviewer_P1F7 · 2022-10-19

**Confidence:** 4
**Correctness:** 1
**Technical Novelty And Significance:** 1
**Empirical Novelty And Significance:** 1
**Recommendation:** 3

**Clarity, Quality, Novelty And Reproducibility:**

Clarity: The language is clear, but the complexity of the method made it unclear to me although, with difficulty, I think I managed to understand it.

Quality: as mentioned above the key claims are unsupported, ablations and state-of-the-art comparisons are missing. I do not consider the quality as high.

Novelty: While I do not think this mechanism has been proposed before, I did not see new ideas which I could clearly "take home" from this paper.

Reproducibility: The authors claim this can be implemented using a single line of code, however it would have been helpful to see the code.

**Strength And Weaknesses:**

The task of anomaly detection is obviously important. Reconstruction-based methods have been around for a long time, but getting them to SoTA performance would be very interesting.

I am not positive towards this paper. It makes very strong claims but does not deliver.
* It claims "We experimentally demonstrated the human brain’s similarity between RFTM and hippocampus-cortex structures". This is a really strong claims and I do not think there is sufficient evidence to support it.
* It claims "state-of-the-art experimental results" but in fact the experiments show it's ROCAUC on Cifar10 is around 50% (random performance). The comparison was only made against memory-based AE methods which are far from SoTA. Comparison against CSI (Tack et al.) or PANDA (Reiss et al.) would not support the above claims.

Furthermore, the approach is probably too complex for its level of performance. The design choice is not justified by a sufficient ablation study making it unclear which aspects are in fact helpful.
Finally, the complexity of the method made it unclear to me. With difficulty I think I managed to understand it, but it still seems arbitrary to me.

**Summary Of The Paper:**

This paper tackles anomaly detection using a memory-based autoencoder approach. The main problem with reconstruction-based AD is that both normal and anomalous examples are well reconstructed making the anomaly score weak. Memory-based AE approaches have not addressed this problem sufficiently. This paper proposed a fairly complex memory mechanism which claims to overcome those issues. It draws inspiration from human cognition (although the precise analogy is not made very clear). The method is compared to a handful of memory-based AE approaches on simple dataset, and claims to obtain better results.

**Summary Of The Review:**

The paper tackles an important problems but the method is not well motivated, unclear. The experimental results are also unsatisfactory.  I recommend rejection.

#######################################################

I have read the the two responses by the authors but would like to keep my rating.

---

> ### Author Response · Authors · 2022-11-18
> **The review you have given is based on a completely wrong understanding.**
>
> # Respond Overview
>
>  **The review you have given is based on a completely wrong understanding.**
>
> # Respond to your mention of "This is a really strong claims and I do not think there is sufficient evidence to support it."
>
> The human brain’s similarity between RFTM and hippocampus-cortex structures are shown in Table2, Figure5, Figure6, Figure7 and Section4.5. Briefly, there are several similarities as follows. First, the generation of memories is prototypically clear compared to MemAE, which is as abstract but representative as human memory; second, the combination of different sizes of memory capacity brings better performance, which is as small capacity short-term memory in the hippocampus with large capacity long-term memory in the cortex; third, RFTM is two modules in series, which is as human recall memory will pass through the hippocampus and then through the cortex.
>
> # Respond to your mention of "in fact the experiments show it's ROCAUC on Cifar10 is around 50%"
>
> We believe that this response stems from your underestimation of the difficulty of the USAD experimental setup. Our results are 1.31% higher than AE, 0.44% higher than MemAE and 1.27% higher than MNAD on CIFAR10. It is important to note in particular that none of the experiments in the USAD setting have a very satisfactory performance on the CIFAR10 dataset. We call on the machine learning community to make the experimental results of USAD public, especially for CIFAR10.
>
> # Respond to your mention of "Comparison against CSI (Tack et al.) or PANDA (Reiss et al.) would not support the above claims" & "far from SoTA"
>
> **CSI's experiments do not obey USAD.** The CSI test set contains a large number of non-semantic anomalies, which is contrary to USAD. Non-semantic anomalies are samples from other datasets that are treated as anomalies, and such anomalies are easily detected. A semantic anomaly is one that treats a class within the same dataset as an anomaly and the remaining classes as normal.
>
> **The experiments of PANDA do not obey USAD.** The training set of CPANDA is labeled multiclass data, which is contrary to USAD. USAD is an unlabeled multiclass scenario, which lacks self-supervised information compared to a labeled multiclass scenario.
>
> In summary, your comment about "far from SoTA" is based on your _**completely wrong understanding of USAD**_. Semantic anomaly detection in label-free multiclass scenarios, i.e. USAD, is a new and challenging research area. USAD needs to be properly understood in order to objectively understand the specific work and contributions of this paper.
>
> CSI: Novelty Detection via Contrastive Learning on Distributionally Shifted Instances
>
> PANDA: Adapting Pretrained Features for Anomaly Detection and Segmentation
>
> # Respond to your mention of "the approach is probably too complex"
>
> If there existed structures that were simpler and more elegant than ours, and achieved the same performance in USAD experiments without introducing penalty terms as we did, that would be groundbreaking work that would excite and delight us. However, until a simpler model with the same performance is available, it is intuitive to give a "too complex" opinion as a reviewer without a reference. This is like saying that Transfomer is a too complex model without giving a simpler model as a basis.
>
> # Respond to your mention of ”insufficient ablation study“
>
> As the only reviewer who has questioned the adequacy of the ablation experiments, we value your comments and suggestions. We are very grateful and excited when constructive suggestions are received, and we would appreciate it if you could give us specific ablation experiments that need to be added and specify the reasons why they need to be added. However, we are very confused by your assertion that the ablation experiments are inadequate without providing the constructive content described above.
>
> # Respond to your mention of "the complexity of the method made it unclear to me"
>
> Reviewer 8c7P gives the following two comments. 1."The paper is well written." 2."The paper may be reproducible as implementation and experiment details are both provided." However, the same paper may appear to be a different reading experience due to differences in article writing level and reading comprehension. We are sorry that our method was unclear for you to understand. Once again, we are very grateful and excited to receive constructive comments and suggestions. We would appreciate it if you could provide us with specific passages, diagrams or pictures that you have difficulty understanding, and specify what makes it difficult for you to understand. We will do our best to rework the unclear part to make it as easy to understand as possible for every reader.
>
> # Final Respond
>
> Thank you for providing a macro-evaluation based on the reader's perspective, which is our motivation to keep improving the paper. However, it would be more substantial for us to accompany it with comments on a specific micro level.

---

> ### Comment · Reviewer_P1F7 · 2022-11-21
> **Post-rebuttal**
>
> I am not convinced by the rebuttal.
> * The similarity to the human brain is still vague. The rebuttal is still far from a conclusive demonstration.
> * I also do not believe the authors describe CSI and PANDA correctly. CSI is evaluated in many settings, not the OOD one. PANDA does not use labelled multi-class data. The results on Cifar10 are not SoTA, contrary to the claim of the rebuttal.
> * Clarity was also an issue for other reviewers as well, although not all.
> Overall, I think this paper should be rejected.

---

> > ### Author Response · Authors · 2022-11-21
> > **Thank you for your criticism and correction. It would be appreciated if you could guide us on some confusion.**
> >
> > # Respond Overview
> >
> > Thank you very much for pointing out our mistakes, and we are very sorry that we may have made mistakes in the understanding of the paper, but we have some confusion, and hope to get your guidance.
> >
> > # Respond to your mention of "The similarity to the human brain is still vague."
> >
> > Thank you for your comments that made us aware that some claims may be too strong, and different readers will have different degrees of agreement with claims, which we will carefully consider in future work.
> >
> > # Respond to your mention of "CSI is evaluated in many settings, not the OOD one."
> >
> > We do not deny that there are many experimental Settings for CSI and not just OOD one. We would like to emphasize that CSI does not contain the USAD setting, which is precisely the focus of our study in our paper. The USAD setting is different and more challenging than the settings of the CSI paper. It is unfair to compare our method and CSI to draw the conclusion of "far from SoTA" under different experimental settings.
> >
> > # Respond to your mention of " PANDA does not use labelled multi-class data."
> > ## [A].Does PANDA use unlabeled data?
> >
> > We conclude that PANDA uses labelled multi-class data based on the following text in paper, "Pretraining, like Outlier Exposure, is also achieved through an external **labelled dataset**." Here is the original paragraph, which we copied and pasted from the article link below  https://arxiv.org/pdf/2010.05903.pdf
> >
> > In addition, we think there is unfairness in comparing models based on **pre-training** and models based on **non-pre-training**.
> >
> > ## [B].PANDA does not have an experimental setup for USAD.
> >
> > PANDA does not have an experimental setup based on USAD. The last line of the PANDA abstract claims that "Our method, PANDA, outperforms **the state-of-the-art in the OCC, outlier exposure and anomaly segmentation settings** by large margins", indicating that his results do **NOT** contain the USAD setting.
> >
> > ## [C].We don't think we are "far from SoTA" under the double inequity factor.
> >
> > First, PANDA uses additional pretraining as described in [A]; Second, PANDA's SOTA results do not contain USAD settings as described in [B]. So we don't think we are "far from SoTA" under the double inequity factor.
> >
> > # Respond to your mention of "Clarity was also an issue for other reviewers as well, although not all"
> >
> > Thank you very much for your positive comment about "the language is clear", and your response about "clarity was also an issue for other reviewers as wel, although not all" was very important to us, as it made us realize that we have an obligation to rewrite some parts to ensure that all readers can understand them clearly.
> >
> > # Final Respond
> >
> > Thank you very much for your reply, which made us realize that over-strong claims and unclear statements are the key points of this article that need to be improved. Regarding the CIFAR10 results, we still consider the USAD setting to be an extremely challenging problem without pretraining. The CSI results with respect to CIFAR10 were not performed under the experimental setup presented in this paper. We suspect that the CSI model would not have performed as well if it had not been pre-trained and tested under USAD, and this is what we will focus on in our next work.

---

### Official Review · Reviewer_8c7P · 2022-10-24

**Confidence:** 3
**Correctness:** 4
**Technical Novelty And Significance:** 2
**Empirical Novelty And Significance:** 3
**Recommendation:** 5

**Clarity, Quality, Novelty And Reproducibility:**

The paper is well written. The provided ablation study, illustration, and insights are helpful, too.

The focus of this paper is to design an improved memory-based and bio-inspired learning module. There are similar methods in this lineup already, although the new one improves the performance on USDA benchmarks.

The paper may be reproducible as implementation and experiment details are both provided.


**Strength And Weaknesses:**

Strength:
* The new method is promising compared to other memory mechanism-based anomaly detection methods under the unsupervised semantic anomaly detection (USAD) setting.

Weakness:
* The major issue is about the experiment design that only USAD setting is considered. While this setting is challenging, other routine settings would be considered as well. This allows the new methods to be evaluated on a variety of data such as image/video datasets.
* How about the performance of the new method compared to non-reconstruction methods, e.g., [1], which also provides appealing performance under the USAD setting?
* While it is fair to use the same encoder/decoder in the experiments, readers may be more interested in other methods, e.g., DAAD mentioned in the “introduction.”

[1] Jihoon ,et al; CSI: Novelty Detection via Contrastive Learning on Distributionally Shifted Instances


**Summary Of The Paper:**

The paper mainly explores the issue of “overgeneralization” in unsupervised anomaly detection and proposed a bio-inspired solution, termed “Random Forgetting Twin Memory (RFTM).” The new model did not change the fundamental structure of existing learning models, e.g., autoencoder, and work in a plug-and-play fashion. It achieves state-of-the-art performance on several public benchmarks.

**Summary Of The Review:**

In brief, the new method developed an interesting bio-inspired RFTM model under the USAD setting to solve the issue of “overgeneralization.” The model details are well-elaborated. However, the experiments are inadequate as only the USAD setting is considered. Other routine settings and non-reconstruction methods should be considered and compared in a more comprehensive way.

---

> ### Author Response · Authors · 2022-11-18
> **Thank you for your positive comments and constructive suggestions**
>
> # Respond Overview
>
> Thank you for the positive comments in your review, it means a lot to us.
>
> # Respond to your mention of "While this setting is challenging, other routine settings would be considered as well. "
>
> It should be noted that this paper studies OGP and the problem only arises under USAD, so the routine experimental setup is not relevant to the field of study of this paper. But still, thank you very much, as this is a very constructive suggestion to allay some of the readers' concerns.
>
> # Respond to your mention of "How about the performance of the new method compared to non-reconstruction methods, e.g..."
>
> This is a very good suggestion, but time constraints did not allow us to complete the rewriting of the CSI code under USAD. It should be noted that the CSI results are not achieved under USAD, since his anomaly data have a significant proportion of non-semantic anomalies. However, we will try to contact the authors to jointly promote the prosperity of OGP research under USAD.
>
> # Respond to your mention of "readers may be more interested in other methods, e.g...."
>
> The reason DAAD mentions it in the article is to show that the existing research trend has moved towards using modules rather than improving them. This paper is a structural improvement of the module, just like ResNet. It would be unfair to force a comparison with DAAD in this paper because DAAD has adopted the Unet rather than AE architecture. If the AE architecture is adopted, DAAD degenerates into MemAE. But still, thank you very much for your suggestions that allow us to think from the reader's point of view without the writer's blind spot.
>
> # Final Respond
>
> While the addition of partial comparison experiments allows for a more comprehensive examination of RFTM, the focus of this paper is on addressing OPG, and the generalization of the module experiment is, as you say, interesting, but not main focus given the space constraints. In conclusion, we were fortunate to be assigned a professional reviewer who helped us think from the reader's perspective and revisit the paper with objective and constructive comments. Thank you.

---

### Official Review · Reviewer_c8kV · 2022-10-24

**Confidence:** 5
**Correctness:** 2
**Technical Novelty And Significance:** 1
**Empirical Novelty And Significance:** Not applicable
**Recommendation:** 3

**Clarity, Quality, Novelty And Reproducibility:**

The presentation quality is below the average of ICLR given the issues identified above. The originality of the paper is also weak.

**Strength And Weaknesses:**

Strengths.
- The studied problem - having small reconstruction errors in reconstructing anomalous samples - is a common problem in autoencoder-based AD approaches
- In the presented method, the key idea of enforcing less prototypes in the memory to alleviate the problem is plausible
- A set of empirical results on three commonly used datasets is used to justify the effectiveness of the proposed method.

Weaknesses.
- Although the key idea is plausible, it shares similar insights to the previous memory network-based autoencoder approaches for anomaly detection, such as MNAD. The main objective there is to learn less yet compact prototypes to avoid the overfitting of the data. Although the specific way to achieve this objective is different from each other, the new prototype learning in this work does not make clear major contributions. The proposed method seems to simplify the memory learning, but its effectiveness is not clear (see comments below).
- The presented results are not convincing. 1) The reported results of the competing methods in the paper are significantly worse than the ones in the their original papers and some recent relevant papers [A,B]. 2) The performance of the presented method is far below that of the current SOTA models (e.g., see the results on the three datasets in [A,B]).
- There are a number of false/misleading claims, such as what does it mean by unlabeled normal samples in "Unsupervised anomaly detection (UAD) only needs to fit the unlabeled normal samples to learn the normal patterns" (why is it unlabeled if the samples are known to be normal), and "most of the experiments were not conducted based on the USAD setup" when reviewing memory network-based AD methods.
- It is unclear why the learned memory can be named as Hippocampus and Cortex memory.
- It claims that the method can work as a plug-and-play component, but not empirical results are given to support this claim.

References.
- A. "Anomaly Detection via Reverse Distillation from One-Class Embedding." In Proceedings of the IEEE/CVF Conference on Computer Vision and Pattern Recognition, pp. 9737-9746. 2022.

- B.  "Deep one-class classification via interpolated gaussian descriptor." In Proceedings of the AAAI Conference on Artificial Intelligence, vol. 36, no. 1, pp. 383-392. 2022.

**Summary Of The Paper:**

The work presents a memory network-based autoencoder method for unsupervised anomaly detection, in which random forgetting gates and top-k prototype selection functions are used to regularize the autoencoder networks to avoid overfitting.

**Summary Of The Review:**

The paper has major issues in main claims, empirical justification and clarity, and its technical novelty is marginal.

---

> ### Author Response · Authors · 2022-11-18
> **The review you have given is based on a completely wrong understanding.**
>
> # Respond Overview
>
> _**The review you have given is based on a completely wrong understanding.**_
>
>
> # Resepond to your mention of "similar to previous work"
>
> Prototype learning and memory learning are related work in this field. I am glad that you have summarized the commonalities in the field at the macro level. Three points to note are the following. First we did not introduce a penalty term in the loss function, which has never appeared in previous work. Secondly the structure we propose is a tandem structure, which has never appeared in previous work. Thirdly we deal with the minimum unit of the instance feature, not the part of the instance feature, which has never appeared in previous work. We believe that the innovation in structure is not a so-called similarity to previous work, but an elegant and simple innovation like ResNet.
>
> # Resepond to your mention of "The presented results are not convincing 1)..."
>
> I am sorry that you received exactly the opposite understanding on the experimental setup. Just like the papers [A,B] you provided, their experimental setup is unlabeled single-class while our article's experimental setup is unlabeled multi-class, which is why the title of the articles [A,B] have "One-Class" in them.
>
> # Resepond to your mention of "The presented results are not convincing 2)..."
>
> You provided experimental results for unlabeled single-class anomaly detection, while our paper shows experimental results for unlabeled multi-class anomaly detection.  Your so-called "far below" is a wrong conclusion based on a completely different experimental setup. It should be additionally noted that the difficulty of unlabeled multiclass anomaly detection is much higher than that of unlabeled single-class anomaly detection.
>
> # Resepond to your mention of "There are a number of false/misleading claims"
>
> **Your comment is based on a complete misunderstanding.** In this paper, we propose a label-free multiclass scenario, which implies that normal data has more than one class, which is the main part that distinguishes it from the previous work. This is why we claimed that "most of the experiments were not conducted based on the USAD setup".
>
> # Resepond to your mention of "why is it unlabeled if the samples are known to be normal"
>
> Because normal data for anomaly detection may be multi-class labeled and multi-class unlabeled, we are obliged to clarify in order for the reader to understand our experimental setup.
>
> # Respond to your mention of "It is unclear why the learned memory can be named as..."
>
> The reasons for naming the hippocampus and the cortex are explained in our original abstract. First, because our idea of connecting the two components in series was inspired by the structure of the hippocampal and cortical cascades. Second, the memory twin has a better performance at different capacities, which is similar to the small capacity short-term memory of the hippocampus and the large capacity long-term memory of the cortex.
>
> # Respond to your mention of "It claims that the method can work as a plug-and-play component, but..."
>
> We open source the code, but releasing the github link in a double-blind review would give away author information. As reviewer 8c7P stated "The paper may be reproducible as implementation and experiment details are both provided", we stated all details in the paper and we point out that plug-and-play can be achieved with a wrapper module.
>
> # Final Respond
>
> A more deliberate and detailed reading needs to be done anew.
>
> A more objective and more respectful review for the article's work needs to be re-released.

---

### Official Review · Reviewer_ZPFH · 2022-10-25

**Confidence:** 3
**Correctness:** 3
**Technical Novelty And Significance:** 2
**Empirical Novelty And Significance:** 2
**Recommendation:** 5

**Clarity, Quality, Novelty And Reproducibility:**

Clarity: not clear why density would be needed in the space to avoid overgeneralization. Not clear the figures of Fig.7, and how they demonstrate the goodness of the proposed approach.

Quality: Results are obtained wrt a limited number of state of the art approaches.

Novelty: the approach seems novel, but I cannot judge how much powerful given the limited comparisons

Originality: Cortex and hippocampus parallelism with the medicine should be more stressed since they are very attractive as justification of the approach.

**Strength And Weaknesses:**

Strenghts:
Unsupervised anomaly detection is a very interesting topic, more insidious than outlier detection.


Weaknesses:
The comparative results are obviously insufficient, since only Vanilla AE, MemAE and MNAD were used as comparison models. Instead, Trust-MAE, DAAD, [Ruff et al., 2021] should be considered (and already mentioned by the authors in the state of the art). Other approaches are

-- Deep Unsupervised Image Anomaly Detection: An Information Theoretic Framework 2020
-- FastFlow: Unsupervised Anomaly Detection and Localization via 2D Normalizing Flows
-- Self-Supervised Predictive Convolutional Attentive Block for Anomaly Detection
--Robust Subspace Recovery Layer for Unsupervised Anomaly Detection  2019
These approaches should also be present and discussed in the state of the art

Second, the study of figure 7 is not clear at all. The space constructed by the autoencoder and the two types of memory of the RFTM should have cavities (autoencoder) and less cavities (the second type of memory of the RFTM in particular). This is somewhat visible in the figures b and c. What is not clear is why the model needs to have a representation space that is dense enough that the anomalies are always forced to be embedded in the distribution of normal representations. Isn’t this a way to hide anomalies instead of highlighting them?


**Summary Of The Paper:**

The model consists of three parts: encoder, decod er, and RFTM, The encoder
makes a nonlinear transform to the input data x in the latent space. The latent data goes through a re-representation (in the same dimensionality) by the RFTM. Then the data is reconstructed by the decoder. The anomaly of the test sample is measured by reconstruction error. In particular, the RFTM  uses limited prototypes to record and re-represent the latent representation, which makes suppression of overgeneralization feasible, and can be trained end-to-end without introducing any additional penalty terms on top of the original task loss function. The rest of the paper is used to show the functioning of the RFTM module with some comparative results and a lot of ablation studies.


**Summary Of The Review:**

- state of the art is insufficien, and I have provided some of the SOTA approaches for UAD
- qualitative results in FIg.7 are not clear

---

> ### Author Response · Authors · 2022-11-18
> **Adequacy statements and correction of misunderstandings**
>
> # Response Overview
>
> Thank you very much for reviewing this paper in your busy schedule, we know it is a very laborious task. Your comments have been very constructive and helpful, and we are happy to have received some of your comments and suggestions on our work. We have provided three targeted responses to the three questions you raised. We also launched a call for the machine learning community to focus on the area of this paper and build it together.
>
> # Response to your mention of inadequate comparisons
>
> First, each row of results is an independent experiment in the experimental setup of this paper, and the mean value is only used as a simplified comparison. Second, we conducted a total of 120 independent experiments with four algorithms on three datasets, and the actual number of experiments was 360 considering three runs taken as average. Third, the purpose of mentioning Trust-MAE and DAAD in this paper is to illustrate that the current existing research on memory modules has gone down two representative paths, which is different from and the highlight of this paper. The former is a more complex loss function penalty term (6 penalty terms) and the latter is a more complex network structure (memory modules placed at each layer). Existing studies have stopped trying to develop new memory modules and instead try to introduce more penalty terms or improve more complex model structures. Fourth, due to the complexity of Trust-MAE and DAAD combined with the absence of open source code or even unofficial implementation code, the results are not convincing even if we reproduce them for comparison. Fifth, the network structure of DAAD cannot be compared fairly in this paper, and MemAE is a degenerate form of DAAD if a fair comparison is forced.
>
> # Response to your mention of should-also-be-present-and-discussed approaches
>
> Of the four papers you mentioned, A and B do not have open source or unofficial implementation code. For paper C, we have performed a comparative reproduction and **still maintain the state of the art** as shown in Table1. For paper D, D requires that the bottleneck of the autoencoder needs to be set to 10, which is obviously an unfair comparison since we require all comparison algorithms to be performed under a 128-dimensional bottleneck model to verify the suppression of OGP.
>
> |         |  Paper[D]  |    **Ours**    |
> |:-------:|:----------:|:--------------:|
> |  MNIST  | 63.56±0.41 | **83.83±0.86** |
> | Fashion | 66.75±0.42 | **71.67±0.58** |
> | CIFAR10 | 50.35±0.47 | **51.48±0.20** |
> Table1.Average AUROC Results
>
> [A]Deep Unsupervised Image Anomaly Detection: An Information Theoretic Framework
>
> [B]FastFlow: Unsupervised Anomaly Detection and Localization via 2D Normalizing Flows
>
> [C]Self-Supervised Predictive Convolutional Attentive Block for Anomaly Detection
>
> [D]Robust Subspace Recovery Layer for Unsupervised Anomaly Detection
>
> # Response to your mention of "why the model needs to have a representation space that is dense enough...?"
>
> The essence of the significant reconstruction error of anomalous samples originates from the failed reconstruction of the autoencoder. So in order to create failed reconstructions, it is necessary to create failed representations due to the fact that the reconstructions come from the decoding of the representations by the decoder. If cavities are present, anomalous samples are mapped to the cavity part by the encoder and then reconstructed by the decoder. Since the cavity part encodes the anomaly information so the reconstruction is successful and the reconstruction error is undetectable. If the cavity is not present, the anomalous sample is mapped to the normal representation by the encoder and reconstructed by the decoder. In this case the decoder will output some normal sample, but it must be present in the training set. Thus, it can cause a failed reconstruction. At this point the reconstruction error becomes large and the abnormal sample is able to be detected. This is the reason why the representation space should be as dense as possible without leaving cavities for anomalous representations.
>
> # Call for Comparison
>
> We call on the machine learning community to conduct experiments on MNIST, Fashion, and CIFAR10 for the USAD experimental setup of this paper, and would be happy to conduct public comparisons to promote progress in the research area covered in this paper. It would be exciting to have work that outperforms the performance mentioned in this paper for the same fair comparison.
>
> # Final Response
> We thank you for pointing out some of the issues, and we have responded to each of them in the paragraphs above.  For experimental adequacy we have provided an explanation of adequacy and additional experimental comparisons. For Figure 7 it should be clarified that our statement is correct and your understanding is misunderstood. A correct understanding needs to be based on an understanding of the OGP. Finally, thank you for originality suggestion.

---

> > ### Author Response · Authors · 2022-11-18
> > **Correction of errors in the table header of Table 1 in Comment**
> >
> > |         |  Paper[C]  |    **Ours**    |
> > |:-------:|:----------:|:--------------:|
> > |  MNIST  | 63.56±0.41 | **83.83±0.86** |
> > | Fashion | 66.75±0.42 | **71.67±0.58** |
> > | CIFAR10 | 50.35±0.47 | **51.48±0.20** |
> > Table1.Average AUROC Results
> >
> > ## In the header, it should be paper[C], not paper[D], this is a typo, I hope it will not cause unnecessary misunderstanding.

---

### Decision · Program_Chairs · 2023-01-20

**Decision:**

Reject

**Justification For Why Not Higher Score:**

It is clear the paper is not yet ready for publication. See discussion above.


**Justification For Why Not Lower Score:**

N/A

**Metareview: Summary, Strengths And Weaknesses:**

The paper deals with unsupervised anomaly detection by proposing an autoencoder variant that is inspired by the cascade structure in the human hippocampus. The proposed autoencoder, which can be cast within a recent line of research about memory-augmented neural autoencoders, has been evaluated in a number of unsupervised anomaly detection tasks on simple image datasets (MNIST, FashionMNIST, CIFAR10).

The reviewers appreciated the potential direction of the work but highlighted a number of weaknesses in terms of presentation and experimental results. First, the presentation of the contribution suffers from over-claiming results in terms of being better than competitors and its biological plausibility. Second, this is linked to the limited experimental setting which (despite a large grid search of hyperparameters), focuses on simple datasets and provides as a justification of goodness and biological plausibility only accuracy metrics in a reconstruction task. Third, some notable competitors are missing (see reviewers' comments).
The authors' rebuttal did not manage to convince the reviewers as the questions about experimental settings and claims did not receive satisfying answers nor the paper presentation was revised accordingly.

The paper is therefore rejected.

We encourage authors to revise the current version of the manuscript by i) refactoring presentation as to tune-down the biologically-inspired claims and terminology, ii) revising presentation  and ii) extending the experimental comparison as to include ii.a) the same setting as previously introduced memory-augmented autoencoders, ii.b) extensive ablation tests on all experimental settings and iii.c) a controlled (and even synthetic) experimental setting where to validate the claims about the need for the memory augmentation.

**Summary Of Ac-Reviewer Meeting:**

N/A